

# Mispointing correction methods for the conically scanning WIVERN Doppler radar

Filippo Emilio Scarsi[1,2], Alessandro Battaglia[1,3,4], Frederic Tridon[1], Paolo Martire[1], Ranvir Dhillon[3], and Anthony Illingworth[5]

[1]Politecnico of Torino, Torino, Italy
[2]University School for Advanced Studies IUSS Pavia, Pavia, Italy
[3]University of Leicester, Leicester, UK
[4]National Centre for Earth Observation, Leicester, UK
[5]University of Reading, Reading, UK

**Correspondence:** Filippo Emilio Scarsi
filippo.scarsi@polito.it

**Abstract.** Global measurements of horizontal winds in cloud and precipitating systems represent a gap in the global observing system. The WIVERN mission, one of the four candidates to be the ESA's Earth Explorer 11 mission, aims at filling this gap based on a conically scanning W-band Doppler radar instrument. The determination of the antenna boresight mispointing angles and the impact of their uncertainty on the line of sight velocities is critical to achieve the mission requirements. While

substantial industrial efforts are on their ways for achieving accurate determination of the pointing, alternative (external) calibration approaches are currently under scrutiny. This work discusses four methods applicable to the WIVERN radar that can be used to correct antenna mispointing both in the azimuthal and in the elevation directions at different time scales.

Results show that elevation mispointing is well corrected at very short time scales by monitoring the range at which the surface peak occurs. Azimuthal mispointing is harder but can be tackled by using the expected profiles of the non-moving

surface Doppler. Biases in pointing at longer time scales can be monitored by using well established reference database (e.g. ECMWF reanalysis) or ad-hoc ground based calibrators.

Although tailored to the WIVERN mission, the proposed methodologies can be extended to other Doppler concepts featuring conically scanning or slant viewing Doppler systems.

# 1 Introduction

Global observations of horizontal winds are of great scientific importance and have huge economic impact (Stoffelen et al., 2020). This has been widely demonstrated by the ESA Aeolus mission (Stoffelen et al., 2005), the first ever satellite mission to deliver profiles of Earth's wind in the lowermost 30 km of the atmosphere on a global scale, via the measurement of the Doppler shifts in the ALADIN ultraviolet lidar backscattered signals (Lux et al., 2021). Though Aelous contributes less than 1%



inputs in NWP, it significantly improves short-range forecasts as confirmed by observations sensitive to temperature, wind and
      humidity (Rennie et al., 2021). Scatterometer measurements can also provide complementary observations at the surface, with
      significant progress being recently achieved even in presence of strong winds near heavy rain (Polverari et al., 2022). However,
      apart from sporadic and sparse radio soundings and aircraft penetrations, no wind observations are currently available inside
      thick clouds and precipitating systems. Recent advances in data assimilation systems have demonstrated that the dynamical
state of the atmosphere can be inferred not only from clear sky temperature and relative humidity observations but also from
      observations in presence of clouds and precipitation (Geer et al., 2017). Doppler cloud radars have the potential to complement,
      in cloudy conditions (with clouds covering roughly 30% of the tropospheric volume), Doppler lidar observations in clear-sky
      and thin clouds. To fulfil this goal, Illingworth et al. (2018) proposed the WInd VElocity Radar Nephoscope (WIVERN,
      www.wivern.polito.it) hinged upon a single instrument: a conically scanning W-band polarization diversity Doppler radar. The
concept is currently undergoing Phase 0 studies as part of the ESA Earth Explorer 11 selection program.

      Doppler radar measurements from low Earth orbit satellites are challenging (Tanelli et al., 2002; Battaglia et al., 2020;
      Kollias et al., 2022). In fact, the large spacecraft velocity has a threefold repercussion.

1.    In combination with a finite antenna beamwidth, it causes "satellite Doppler fading", i.e. a broadening of the Doppler
      spectrum, which is a synonym of a decreased medium correlation time (Kobayashi et al., 2002). This generally increases
the uncertainties in the Doppler estimates performed with radar pulse pair estimators (Doviak and Zrnić, 1993). In order
      to mitigate this issue, techniques based on polarization diversity (Battaglia et al., 2013; Wolde et al., 2019) and Displaced
      Phase Center Antenna (Durden et al. (2007); Tanelli et al. (2016); Kollias et al. (2022)) concepts have been proposed.

2.    In presence of inhomogeneity within the radar backscattering volume, it introduces non uniform beam filling biases
      (Tanelli et al., 2002; Kollias et al., 2014; Sy et al., 2014).

3.    It requires a very precise and accurate knowledge of the antenna pointing to enable the subtraction of the non-geophysical
      component of the satellite velocity along the line of sight (Tanelli et al., 2004; Battaglia and Kollias, 2014).

      The different source of errors involved with a conically scanning Doppler radar with polarization diversity, as adopted for
      WIVERN, have been thoroughly discussed in Battaglia et al. (2018, 2022) and Rizik et al. (2023). End to end simulations
      suggest that the WIVERN mission requirements on horizontal-projected line of sight winds, of a total random (systematic)
error lower than 3 (0.5) m/s at an integration distance of 20 km for reflectivities above -15 dBZ, are at reach. Such accuracy
      is expected to be sufficient to ensure significant impact in operational NWP, but in previous error budget studies, antenna mis-
      pointing errors have been considered negligible (i.e. < 0.3 m/s both for bias and random error, e.g. see Fig. 11 in Battaglia
      et al. (2022)). Phase 0 industry studies suggests that the mispointing power spectral density (PSD) has indeed larger compo-
      nents than previously expected, particularly for very slow varying components. Thus, methodologies to correct for antenna
mispointing are highly desired in the context of the WIVERN mission and in general for scanning atmospheric Doppler radars.
      In this paper, after introducing the geometry of observation and the impact of mispointing errors on line of sight (LOS) Doppler
      velocities (Sect. 2), different Doppler correction methods are proposed and reviewed, discussing their potential for reducing
      the mispointing errors (Sect. 3). Conclusions and future work are outlined in Sect. 4.



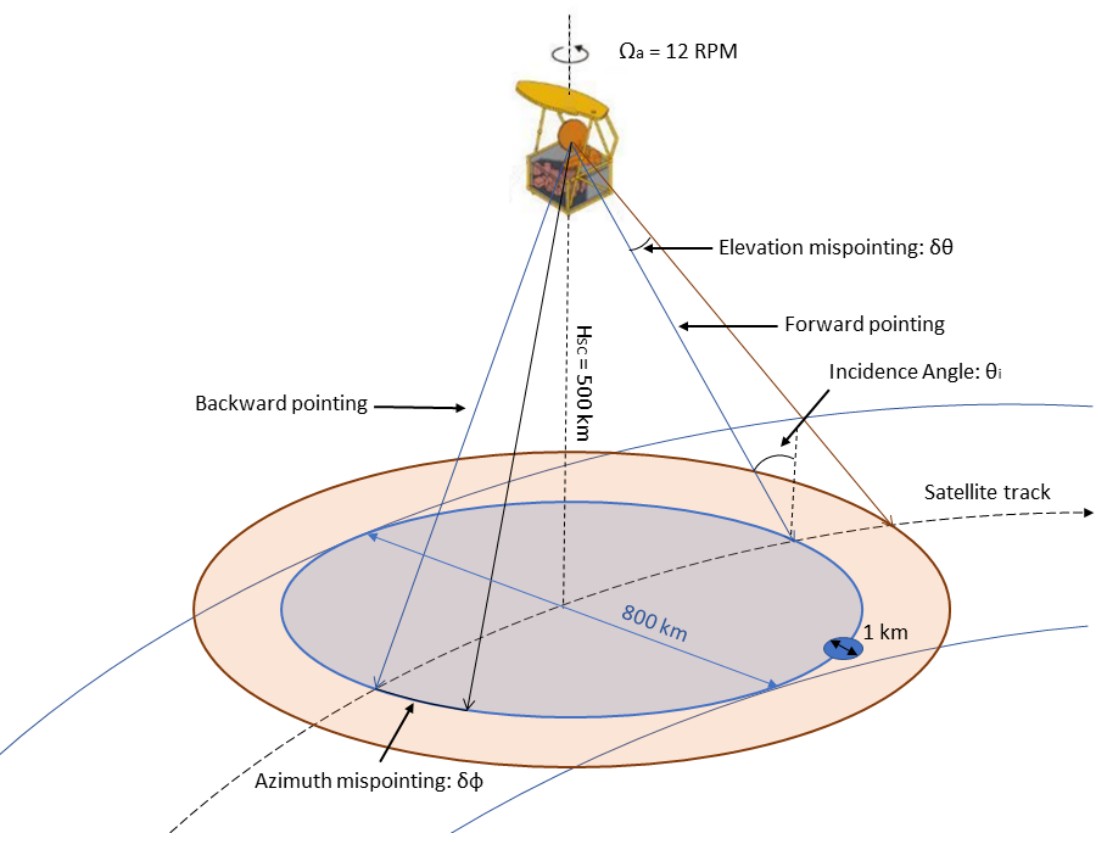

**Figure 1.** Geometry of observation of the WIVERN conically scanning radar: the antenna boresight, indicated by a thin blue line, is rotating at 12 RPM and pointing at a nominal incidence angle of about $42°$. The orange and the black arrows represent an elevation and an azimuthal mispointing in correspondence to the forward and backward pointing configuration, respectively.

## 2 Mispointing errors

Fig. 1 depicts the geometry of observation of the WIVERN radar, whose specifics are listed in Tab. 1. Mispointing in the knowledge of the antenna boresight produces error in the estimates of the hydrometeor Doppler velocity because the component of the spacecraft (S/C) velocity, $v_{S/C}$, along the antenna boresight needs to be subtracted from the measured Doppler velocity. If the actual pointing of the antenna has a mispointing of $\delta\theta$ in the elevation angle and of $\delta\phi$ in the azimuthal, then the mispointing error will be:

$$
\begin{aligned}
\frac{\delta v_{mis}}{v_{S/C}} &= [\sin(\theta + \delta\theta)\cos(\phi + \delta\phi) - \sin(\theta)\cos(\phi)] = [(\sin(\theta) + \cos(\theta)\delta\theta)(\cos(\phi) - \sin(\phi)\delta\phi)) - \sin(\theta)\cos(\phi)] \\
\delta v_{mis} &= v_{S/C}[-\sin(\theta)\sin(\phi)\delta\phi + \cos(\theta)\cos(\phi)\delta\theta]
\end{aligned}
\tag{1}
$$






where the azimuthal scanning angle is assumed $0°$ in the forward direction. Eq. 1 implies that the error in the LOS velocity is modulated by the azimuthal scan frequency since $\phi = \Omega_a t + \phi_0$, where $\Omega_a = 1.26$ rad/s is the antenna angular velocity. Note that a $100\,\mu$rad error in azimuth (elevation) produces a maximum error of 0.5 (0.57) m/s when looking sideways, i.e. $\phi = \pm\pi/2$ (forward/backward direction, i.e. $\phi = 0, \pi$).

## 3   Doppler correction methods

Four different methods have been identified that can be used to correct Doppler errors. Some of these techniques are applicable both to azimuth and elevation mispointing, others only for one type of mispointing; the first two are effective on short time scales (of the order of few ms), the latter two on much longer time scales (weeks/months).

### 3.1   Correction method I: altimeter mode technique

Because of its peculiar illumination geometry, pulse shape and the receiver IF filter response (Meneghini and Kozu (1990)), the WIVERN radar will produce a very specific reflectivity shape for flat surfaces in absence of low level clouds and/or precipitation, with a peak corresponding to the surface range along the boresight direction (Battaglia et al. (2017); Illingworth et al. (2020)). Since the position of the S/C is extremely well known, any discrepancy $\delta z$ between the range of the surface from the S/C computed along the (AOCS-estimated) boresight direction and the measured range of the surface peak can be attributed to an elevation mispointing, $\delta\theta$, from:

$$\delta\theta = \frac{\delta z}{r_{S/C}\,\sin\theta_i} \approx \frac{\delta z\,\cos\theta_i}{\sin\theta_i\,H_{S/C}} \tag{2}$$

where $r_{S/C}$ is the distance between the surface and the spacecraft along the boresight. With the WIVERN specifics (see Tab. 1), a $\delta z = 10$ m corresponds to about 22.5 $\mu$rad. Therefore, this method is not viable for tackling the azimuth mispointing, but has the advantage that it depends only on the reflectivity profile, thus it has the same sensitivity in each part of the scan.

In order to understand the uncertainties associated with this method, realistic surface returns as detected by WIVERN have been simulated starting from the expected flat surface return shape derived following Meneghini and Kozu (1990) (Fig. 5 in Illingworth et al. (2020); an example of a surface return is provided in Fig. 2, black line). The profiles have been scaled in order to produce different Peak to Noise Ratios (PNR), defined as the ratio between the peak reflectivity of the surface profile and the single pulse sensitivity or reflectivity equivalent noise level of the radar, assumed equal to -15 dBZ. For each PNR and each integration length (with 8 independent pulses per km), different stochastic realizations of the signal plus the noise are simulated following the technique proposed by Zrnic (1975). One of such possible realizations is shown as the blue line in Fig.2. Then the signal is noise subtracted and sampled at the WIVERN sampling rate (100 m along range, see Tab. 1) with random range offsets in order to account for the variability of the digitization process along the orbit (red diamonds in Fig. 2). A surface detection criterion is introduced. For each profile, only points which are 3 dB above the detection level (red diamonds with black points inside) are considered. If the profile contains at least 10 consecutive points above the detection threshold is then used for fitting the surface return shape with two free parameters: the peak height and the peak amplitude. The fit is performed via a least mean



**Table 1.** Specifics of the WIVERN radar. The configuration here adopted is the one currently under Phase-0 study for the ESA Earth Explorer 11 program.

| | |
|---|---|
| Spacecraft height, $H_{S/C}$ | 500 km |
| Spacecraft velocity, $v_{S/C}$ | 7600 ms$^{-1}$ |
| Off-nadir pointing angle | 38° |
| Incidence angle, $\theta_i$ | 41.6° |
| RF output frequency | 94.05 GHz |
| Transmitted power $P_t$ | 2 kW |
| Pulse width $\tau$ | 3.3 $\mu$s |
| Antenna beamwidth, $\theta_{3dB}$ | 1200 $\mu$rad |
| Circular antenna diameter | 3 m |
| Antenna angular velocity, $\Omega_a$ | 12 rpm |
| Footprint speed | 500 kms$^{-1}$ |
| Transmit polarization | H or V |
| Cross-polar isolation | <-25 dB |
| Single pulse sensitivity | -15 dBZ |
| H-V Pair Repetition Frequency | 4 kHz |
| Range sampling distance (rate) | 100 m (1.5 MHz) |
| Number of H-V Pairs per 1 km integration length | 8 |

square procedure where each point is weighted by the expected error (errorbars in Fig. 2) computed according to the signal to noise ratio (SNR, following Eq. 16 in Battaglia et al. (2022)). The best fitting curve (blue line) differs from the actual profile for a shift in height, $\delta z$, and a shift in amplitude, $\delta Z$. The variability of $\delta z$ can be statistically retrieved as a function of the PNR by generating a sufficient number of profile realizations with different random noise and different digitalization of the signal. While, as expected (not shown), the mean value of $\delta z$ over different digitization and stochastic noise is practically close to 0 m for every PNR and for every integration length, its standard deviation, $\sigma_{\delta z}$, decreases when either the PNR and/or the integration length increase, as shown in Fig. 3. The shifts in elevation angle $\delta\theta$ corresponding to the shifting in altitude $\delta z$ of the peak are reported on the y-axis on the right hand side of Fig. 3. For instance, with a PNR of 10 dB, the surface position is expected to be determined with an error of about 32, 20, 13 and 9 m for integration lengths of 1, 2, 5, 10 km, respectively. These values correspond to an uncertainty in elevation of 75, 47, 39 and 20 $\mu$rad. The last three solutions guarantee that the velocity error induced by such mispointing will always remain below 0.3 m/s.

### 3.1.1 Statistics of useful surface return

Estimates of the frequency of surfaces exceeding threshold values of PNR have been obtained by exploiting the climatology gathered by the polar-orbiting nadir-pointing CloudSat W-band radar and simulating the WIVERN returns at slant incidence



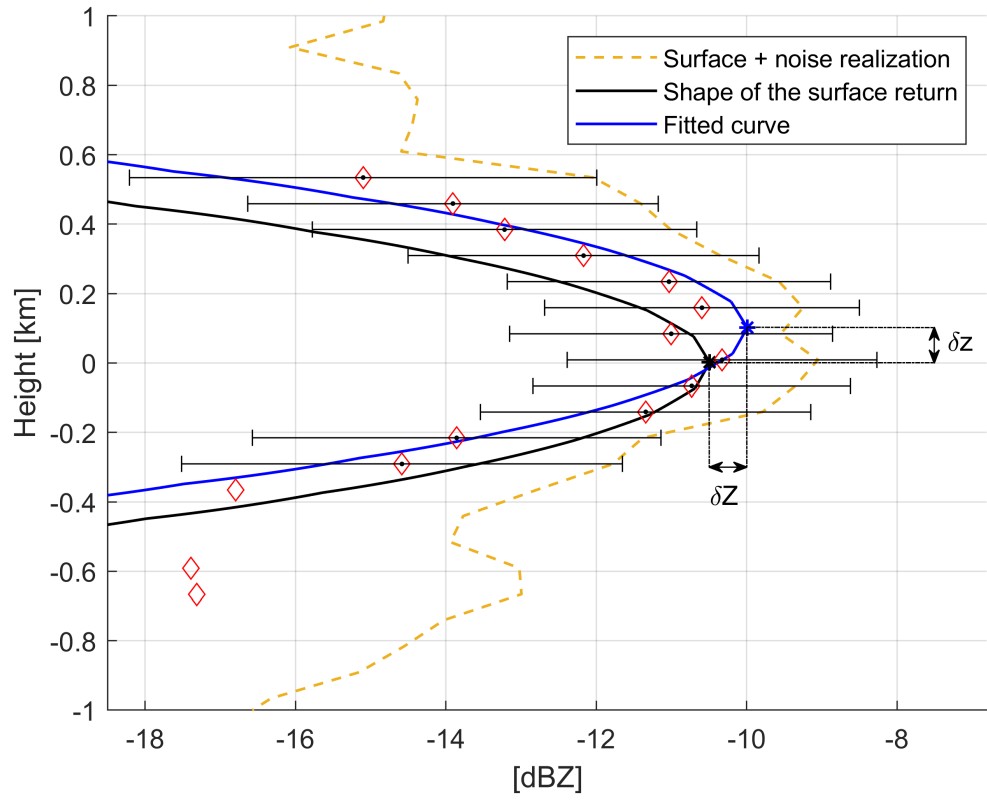

**Figure 2.** A reflectivity surface profile simulated as observed by the WIVERN radar with the procedure to determine the vertical displacement $\delta z$ of the peak of the noisy sampled profile with respect to the actual surface peak. The black line represents the ideal shape of the surface return for a 7 dB PNR with its peak highlighted by a black star. The yellow line represents a stochastic realization of the same surface return with a -15 dBZ random noise. The red diamonds represent the digitized signal after noise subtraction with the error bars indicating the expected errors in the reflectivity estimates. The black dots insides the red diamonds correspond to the points sampled by the radar that are used for fitting the surface profile. The blue line is the best fitting profile, with the peak highlighted by a blue star. The displacements in height ($\delta z$) and in amplitude ($\delta Z$) between the black and blue stars are indicative of the uncertainties associated to the clutter characterization.

angle. The method, proposed by Battaglia et al. (2018) and refined in Tridon et al. (2023), accounts for the additional path integrated attenuation and the reduction in surface normalised backscattering cross sections (Battaglia et al., 2017) when considering the slanted viewing geometry of the WIVERN radar. Fig. 4 shows the cumulative distribution functions (cdf) of the surface peaks (and equivalently of PNR) for land and ocean surfaces. Since ocean surfaces are far less bright than land surfaces at WIVERN incidence angles, a lower percentage of sea than land surface profiles exceeds any given threshold.



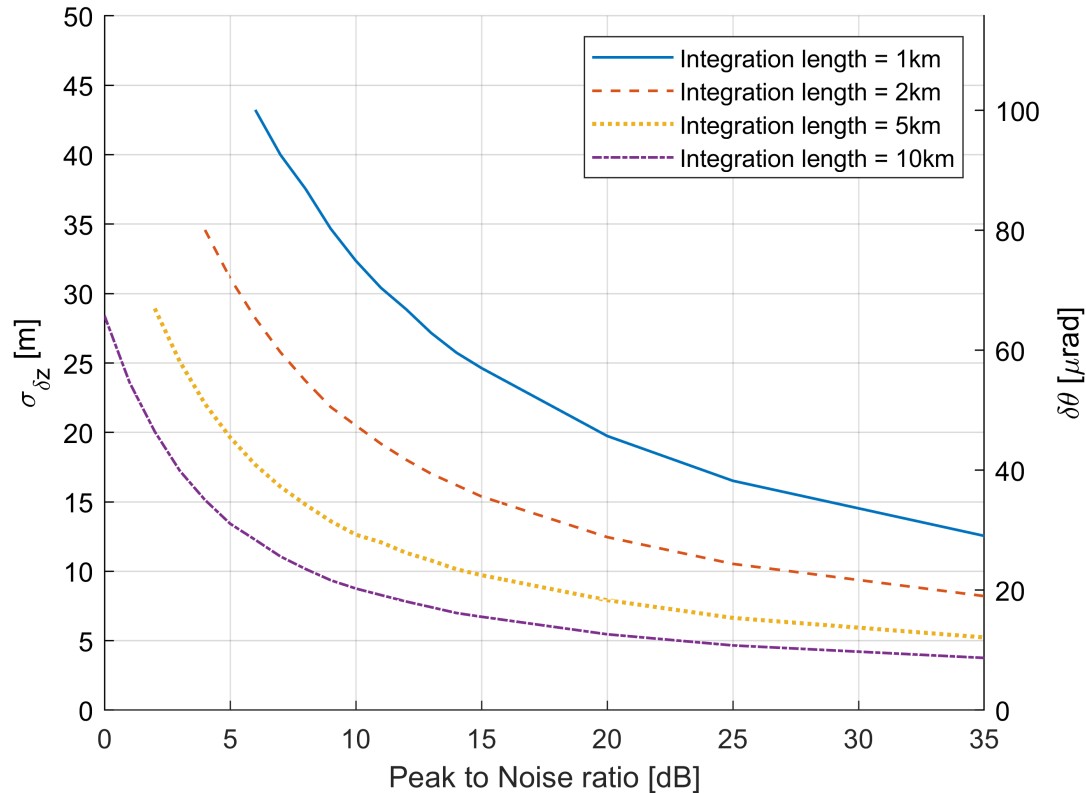

**Figure 3.** Uncertainty in the elevation mispointing determination by the altimeter mode technique: standard deviations of $\delta z$, $\sigma_{\delta z}$ (left axis), and $\delta\theta$ (right axis) as a function of the PNR for different integration lengths, as indicated in the legend. The mean values of $\delta z$ are negligible for all analysed PNR and integration lengths (not shown). The curves are drawn only for PNRs high enough that more than 80% of the profiles satisfy the surface detection criterion. See text for details.

For instance, more than 36% and 99% of the surface peaks exceed -5 dBZ (i.e. PNR=10 dB) for ocean and land surfaces, respectively. Two factors can actually decrease the number of surface returns effectively useful for the calibration:

1. the presence of low clouds and precipitation that could perturb the shape of the reflectivity and of the Doppler profile (discussed later in Sect. 3.2);

2. for land surfaces, the failure of the flat surface assumption.

In order to assess the first issue, we have recomputed the cdf excluding rays where the hydrometeor signal to clutter ratio (SCR) is higher than -20 dB (dotted lines) and -10 dB (dashed lines) in the 750 m closest to the surface. These conditions ensure that the clutter signal is 100 and 10 times stronger than any perturbing signal produced by the hydrometeors. The presence of low



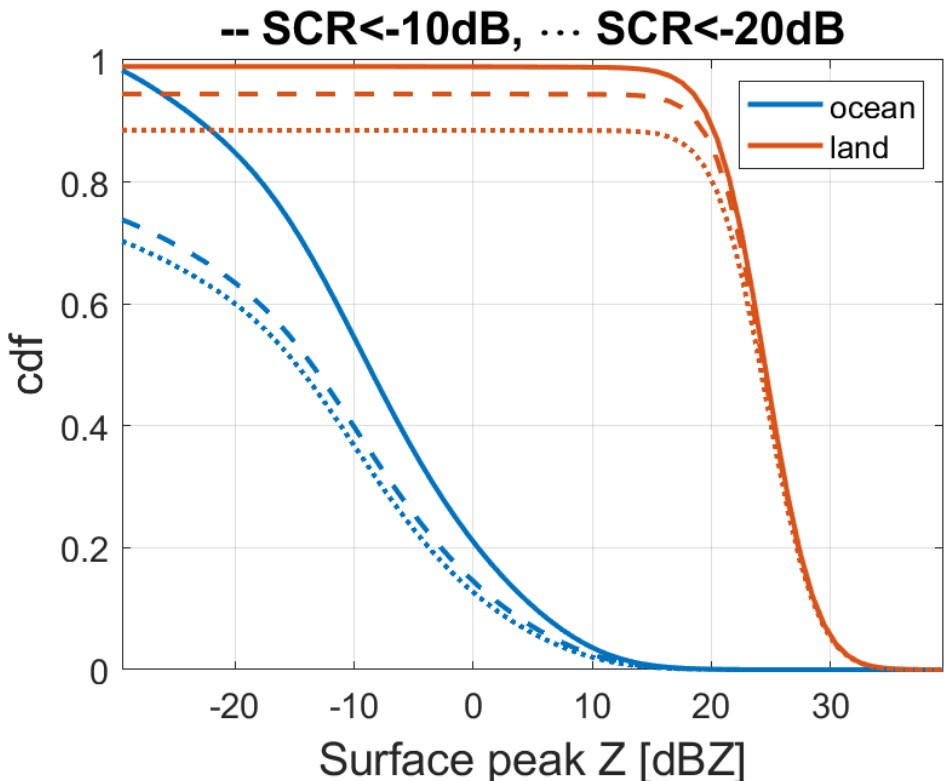

**Figure 4.** Cumulative distributions of surface reflectivity peaks as expected in WIVERN observations for land (red) and ocean (blue) surfaces. Dashed and dotted lines correspond to rays characterised by decreasing signal to clutter ratios (see the text for more details).

clouds will further reduce the number of useful rays for calibration but such reduction is only of 10% for SCR lower than -10 dB and of 13% for SCR lower than -20 dB over ocean. Over land, the reduction is even smaller with 5% and 10% for SCR lower than -10 dB and -20 dB, respectively.

For estimating the impact of the second issue see the discussion in Sect. 3.2.1.

### 3.2 Correction method II: surface Doppler technique

Flat and still surfaces are characterized by a well-determined Doppler profile. But, while the surface reflectivity profiles are independent on the azimuthal scanning angle, the Doppler profiles are azimuthal dependent. Under the assumption of an homogeneous (i.e. backscattering cross section constant across the footprint) flat and still surface, at side view, the surface Doppler is expected to have 0 m/s at all heights whereas, at other azimuthal angles, positive and negative Doppler velocities are expected above and below the surface with 0 m/s only in correspondence to the surface height (black line in Fig. 6). This is 130 the result of the different orientation of the lines of constant Doppler shift induced by the S/C velocity (isodops) with respect



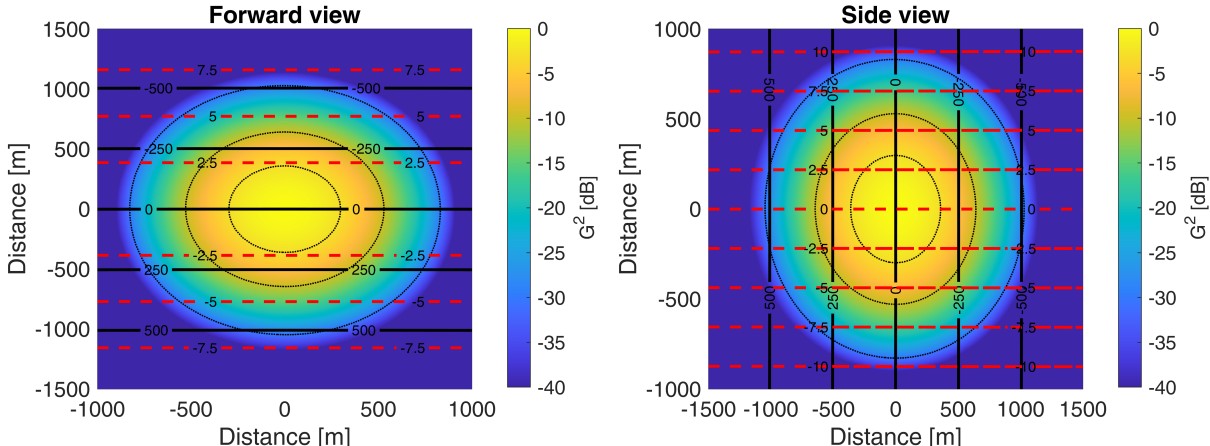

**Figure 5.** Square of the gain (normalised to 0 dB at boresight) for the WIVERN antenna pattern as derived from a previous study (Lori et al. (2017)) for points within 1500 m from the ground projection of the boresight (used as origin of the coordinate system). Results are reported in correspondence to the forward ($\phi = 0$, left panel) and side ($\phi = 90°$, right panel) view. In both cases the satellite is assumed to move along the y-axis. Contour levels of the satellite velocity along the LOS are plotted as dashed red lines from -10 to 10 m/s with 2.5 m/s separation; contour levels of height above the ground are plotted as black lines from -500 to 500 m with 250 m separation. The dotted thin black lines correspond to -3, -10 and -30 dB for the normalised square gain.

to the lines with constant range. As demonstrated in Fig. 5, for the two extreme cases of forward (left panel) and side (right panel) views the isodops are parallel and perpendicular, respectively, to the lines of constant range from the radar.

Like done in Sect. 3.1, the Doppler returns as measured by the WIVERN radar are simulated with the noisiness proper to the given PNR and number of independent samples (according to Eq. (16) in Battaglia et al. (2022), diamonds in Fig. 6). Here,
a perfect correlation between the V and the H pulse co-polar signals is assumed consistently with low surface LDR values (Wolde et al., 2019). The expected shape (black line) is then fitted through the data that have reflectivites exceeding an SNR of 0 dB via a mean least square technique. The distance of the fitted profile (blue line) from the expected profile at zero height ($\delta v_D$) is indicative of the velocity error that can be achieved with this methodology.

Fig. 7 shows the Doppler velocity uncertainties associated with this method. In this case to reduce the error to values lower
than 0.4 m/s is impractical at short integration lengths ($\leq$ 2km) and requires surface with PNR exceeding 10 and 15 dB for integration lengths of 5 and 10 km, respectively. When comparing the curves at different integration lengths, it can be noted that, approximately, the error drops with the square root of the integration length. Therefore this technique seems very promising for correcting mispointing modulated on time scales longer than the antenna rotational period. Surface returns for full rotations in correspondence to clear sky and flat surfaces can be used to fit the mispointing error provided by the expression (1). Because
of the numerous number of good calibration points (i.e. "flat" surfaces in clear sky with goood PNR) expected to be available in different scans this method will constrain mispointing errors down to less than 0.2 m/s within few turns.



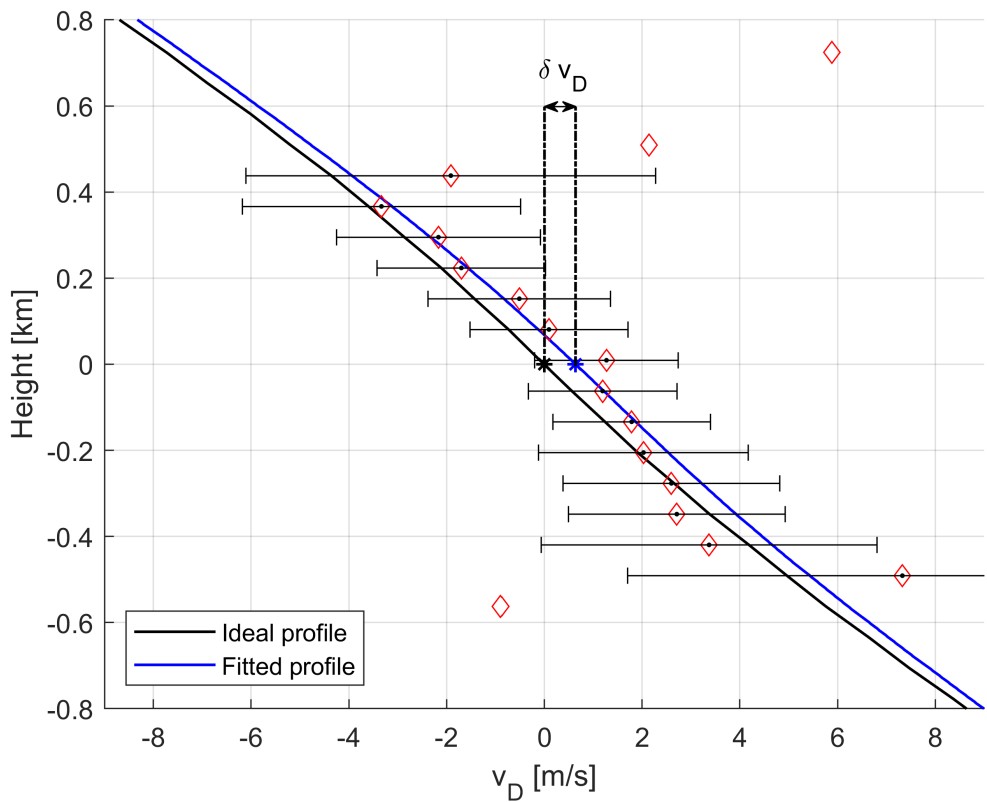

**Figure 6.** A Doppler velocity profile of the surface simulated for WIVERN observations in the forward direction. The black line represents the ideal Doppler velocity profile without the noise. The red diamonds are the points of the noisy Doppler velocity profile that are oversampled by the radar (one point every 100 m along LOS). The black dots identify the points of the noisy profile with reflectivities above the detection level. Such points are fitted with the shape of the surface return in order to produce the blue line. The Doppler velocities represented by the blue line are the ones retrievable from the radar measurements and they differ from the ideal velocities due to the presence of the noise. $\delta v_D$ is the velocity shift in correspondence to the surface induced by the noise in the retrieved profile.

### 3.2.1 Flat surface approximation

The Advanced Spaceborne Thermal Emission and Reflection Radiometer (ASTER) global digital elevation model (DEM) (https://asterweb.jpl.nasa.gov/GDEM.asp), with a resolution of $1'' \times 1''$ (i.e. 30.9 m $\times$ 30.9 m at the equator), has been used to examine the validity of the flat surface assumption by evaluating the variability of the elevation in areas comparable to that swept by the WIVERN radar footprint with different integration lengths. Boxes with latitudinal extent of $18''$ and longitudinal extents of $18''$, $36''$, $72''$, $90''$, and $180''$ were considered. The standard deviations of the elevation, $\sigma_{elev}$, within each box were then calculated for the entire global data set. Fig. 8 shows the variation of the cumulative land fraction with $\sigma_{elev}$, for the five





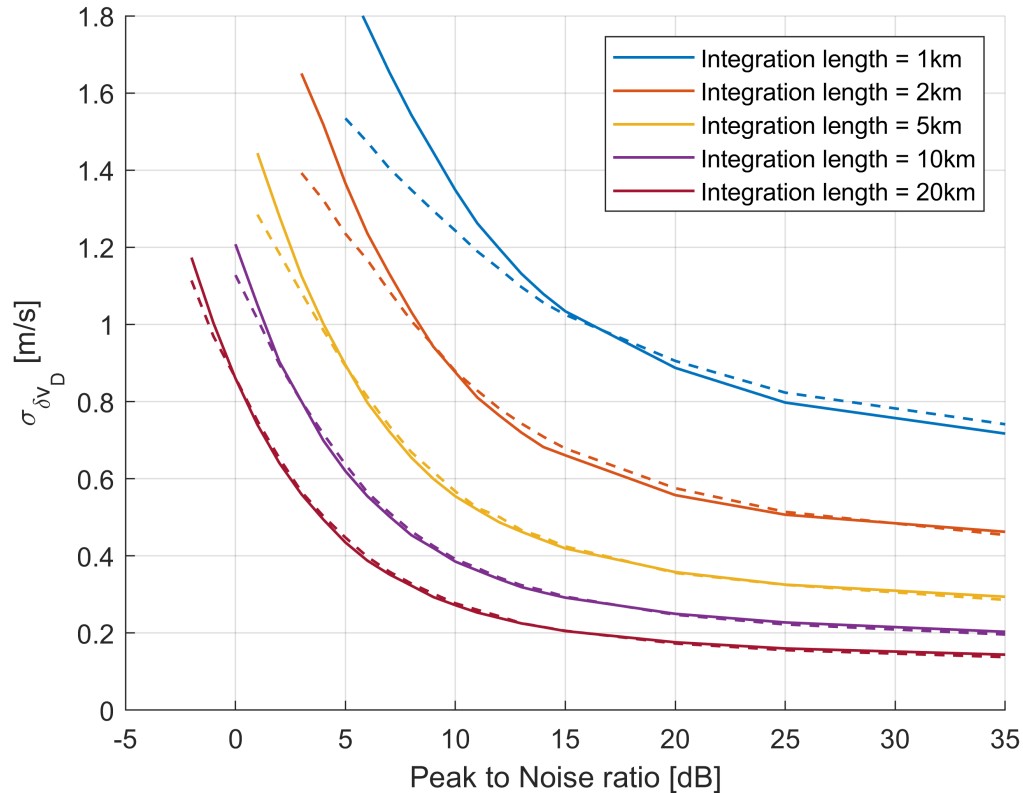

**Figure 7.** Doppler velocity uncertainty associated with the surface Doppler technique for forward pointing (but similar results are found for any azimuthal angle). The standard deviations of $\delta v_D$ is plotted as a function of the PNR for different integration lengths as indicated in the legend. The variability of $\delta v_D$ decreases as the Peak to Noise Ratio and/or the integration length increase. The mean values of $\delta v_D$ are negligible for every PNR and every integration length here analysed (not shown). The curves are drawn only for PNRs high enough that more than 80% of the profiles satisfy the surface detection criterion. See text for details.

box sizes adopted. The respective values of $\sigma_{elev}$ for increasing box size, given in the form ($50^{th}$ percentile, $70^{th}$ percentile, $90^{th}$ percentile), are (7.0 m, 12.0 m, 37.3 m), (8.0 m, 14.4 m, 47.1 m), (9.4 m, 17.9 m, 59.5 m), (10.0 m, 19.2 m, 63.5 m), and (12.0 m, 23.6 m, 74.9 m). It is clear that, for a given percentile, the value of $\sigma_{elev}$ increases as the box size increases. These findings suggest that roughly half of the land surfaces have a variability in elevation less than 10 m within characteristic WIVERN averaging areas. Such surfaces will likely be useful for the calibration methods I-II (Sects. 3.1-3.2), but a dedicated study to properly assess the impact of surface elevation variability is needed.

A final consideration: generally, ocean surfaces will not appear to have zero Doppler velocities but they will have biases of the order of less than 1 m/s. This is due to the interplay between waves and currents (Chapron et al., 2005; Ardhuin et al.,



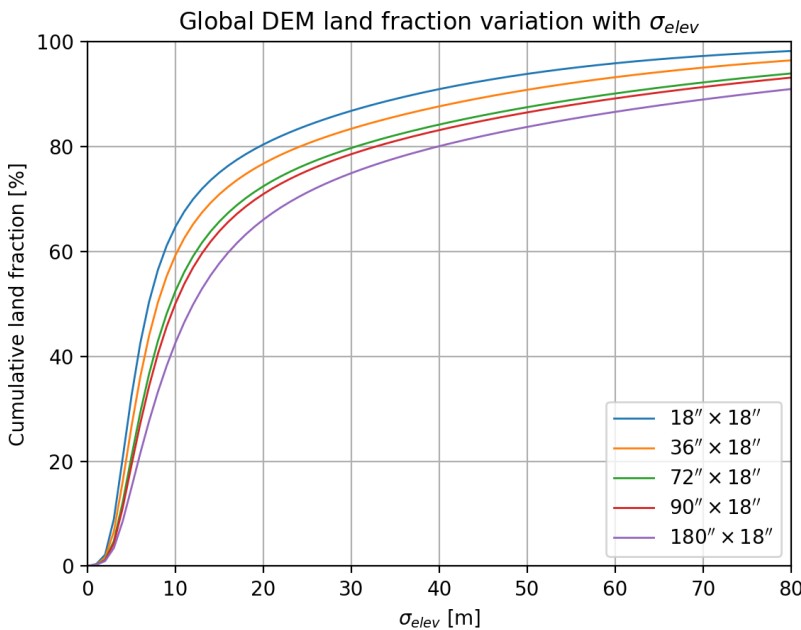

**Figure 8.** Variation of the cumulative land fraction from the ASTER global digital elevation model (DEM) with $\sigma_{elev}$ for different latitude/-longitude box sizes. Each coloured line corresponds to a particular longitude $\times$ latitude box size (indicated in the legend).

2019). Here it is assumed that corrections for such effects can be performed based on auxiliary information or that they will average out when considering looks from different directions.

### 3.3 Correction method III: active radar calibrator techniques

The use of active radar calibrators (ARC) is well established for external calibration of SAR instruments. It has been applied for calibrating the TRMM and GPM radars as well (Masaki et al., 2022).

Here we only consider the use of the ARC in receiver mode with extremely high sampling resolution ($\leq 0.1\mu s$). Like in Masaki et al. (2022) we assume the ARC beamwidth of the order of $20°$, thus much larger than WIVERN beamwidth. Then, in correspondence of an overpass, if the ARC is pointed toward the satellite, the power received by the ARC at the sampling

time $t_k$, $P_{ARC}(t_k)$, is effectively determined only by the WIVERN Tx antenna pattern. Typically, the signal at the ARC will be detectable for few tenth of milliseconds (see right panels in Fig. 9). Since the velocity of the S/C is very low with respect to the scanning velocity of the radar, its effects on the relative motion of the ARC inside the antenna pattern is negligible. Thus, all the possible trajectory of the ARC inside the pattern always look like a slightly bent line extended along the azimuth distance (left panel in Fig. 9).





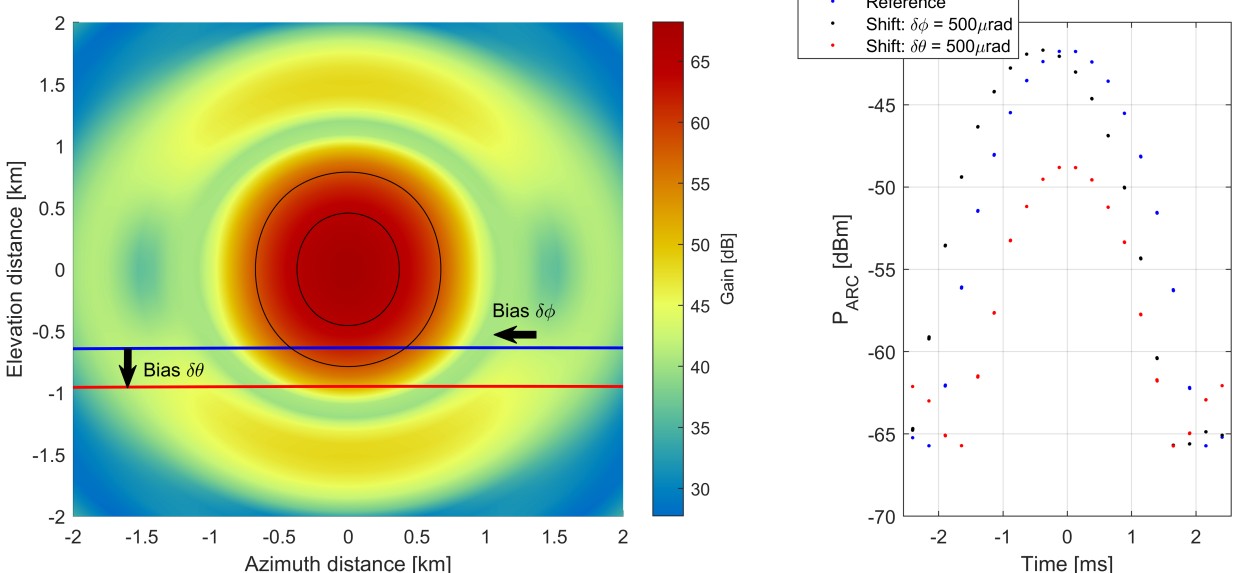

**Figure 9.** Left panel: example of how the ARC position (expressed in terms of distances at the ground) moves inside the WIVERN antenna pattern when a bias in elevation or in azimuth of 500 $\mu$rad is introduced. The satellite is located along the negative y-axis with the antenna pointing forward in the y-direction. The blue line is the position of the ARC for a scanning with a minimum distance between ARC and boresight position of 635 m (reference). The red is the same scan shifted by 500 $\mu$rad in elevation bringing the minimum distance between the ARC and the antenna boresight position to 950 m. Black dashed lines correspond to the contour levels of the antenna gain 3 and 10 dB below the maximum gain. Right figure: the ARC received power for the reference (blue) and the scans shifted by 500 $\mu$rad in elevation (red) and in azimuth (black). The power received is sampled every 0.1 $\mu$s; 3.3$\mu$s pulses are transmitted by the radar every 250 $\mu$s.

Note that the radar receiver chain can also be tested by using the radar as an active calibrator (e.g. by sending back to the radar a copy of the signal received by the ARC). With this technique the power received at the WIVERN receiver will depend on the product of the the antenna gain in receiving and transmitting mode and thus will have a higher sensitivity than the method discussed next.

### 3.3.1    Elevation mispointing

As illustrated in Fig. 9, an elevation mispointing bias moves the apparent motion of the ARC position inside the WIVERN antenna pattern along the elevation direction (y-axis), e.g. from the blue to the red line. Note that uncertainties in the atmospheric refraction between different models (Mangum and Wallace, 2015) are expected to be of the order of 1 arsec (i.e. lower than 5 $\mu$rad) and are therefore neglected in this study. Correspondingly, the actual power measured by the ARC, derived by using Friis formula with an ARC gain of 20 dB, changes because of the difference in the antenna pattern. Such change depends on

the specific position of the overpass (e.g. on the minimum distance of the boresight position to the ARC) and on the details





of the antenna pattern. In order to simulate the capability of the ARC measurement to identify and quantify the mispointing, different possible boresight ground tracks have been simulated (like the blue and the red lines in the left panel of Fig. 9). For each ARC position at a given elevation distance $\bar{\theta}$, the ARC signal is simulated and compared with the returns sampled at different elevation distances $\bar{\theta} + \Delta\theta$ (with a maximum shift $\Delta\theta$ of $\pm 1000\,\mu$rad and sampled every 5 $\mu$rad).

The mean square distances of simulated ARC received signals at position $\bar{\theta}$ and $\bar{\theta} + \Delta\theta$ sampled at different time $t_k$ with $k = 1, \ldots, N_t$ is computed as:

$$MSD(\bar{\theta}, \bar{\theta} + \Delta\theta) = \frac{\sqrt{\sum_{k=1}^{N_t} \left( P_{ARC}^{\bar{\theta}}(t_k) - P_{ARC}^{\bar{\theta}+\Delta\theta}(t_k) \right)^2}}{N_t} \qquad -1000\,\mu rad \leq \Delta\theta \leq 1000\,\mu rad. \qquad (3)$$

All power signals below -80 dBm in the summation have been excluded in order to make sure there will be no effective impact from ARC receiver noise. A (0.5) 1 dB noisiness is introduced in the antenna pattern to account for uncertainties in the antenna pattern. 400 different realizations of the antenna pattern are used to compute the MSD in Eq. (3) so that a distribution of MSD can be derived for each pair $(\bar{\theta}, \bar{\theta} + \Delta\theta)$.

The almost symmetric shape of the antenna pattern causes the similarity between two signals sampled at opposite elevation angles with respect to the boresight (i.e. with $\bar{\theta} = \pm\Delta\theta$). Two examples of the 50th (line) and 5-95th percentiles (shading) of the MSD pdfs are shown in Fig. 10 for $\bar{\theta} = 160\,\mu$rad (left) and $\bar{\theta} = 480\,\mu$rad (right). As expected, the minimum is found in correspondence of a shift between the pairs $\Delta\theta$ equal to 0 $\mu$rad but the interplay between the width of the pdf and the prominence of the local minimum introduces uncertainties in the determination of the mispointing.

When considering an overpass close to the ARC (e.g. $\bar{\theta} = 160\,\mu$rad, left panel), the same $MDS$s are encountered on a vast interval of $\Delta\theta$, ranging inside the first antenna main lobe in a symmetric way, i.e. with $\theta$ between $-\bar{\theta} - \Delta\bar{\theta}$ and $\bar{\theta} + \Delta\bar{\theta}$ with $\Delta\bar{\theta} = 250\mu$rad for this specific example. When considering ARC positions farther away from the boresight, a second local minimum may form in correspondence of $\Delta\theta = -2 \cdot \bar{\theta}$ (e.g., for $\bar{\theta} = 480\,\mu$rad, the second local minimum is at around $\Delta\theta =$ -960 $\mu$rad, right panel).

Left panel of Fig. 11 shows the uncertainty in the elevation mispointing determination as a function of the elevation distance of the ARC from the WIVERN antenna boresight. It is determined by computing for which elevation mispointing the upper 95th percentile, corresponding to the minimum of the MSD (level identified by the red line in Fig. 10), exceeds the lower 5th percentile in the adjacent mispointing angles. The black and the red shading represent the result obtained considering an antenna pattern uncertainty of 0.5 dB and 1.0 dB, respectively. Results show that uncertainty are maximised in correspondence to an overpass of the boresight exactly over the ARC.

Given the fact that there will be multiple overpasses per months (see Fig. 12), it seems realistic to bring this uncertainty down to less than 50 $\mu$rad.

### 3.3.2 Azimuthal mispointing

On the other hand, a mispointing in azimuth does not move the apparent motion of the ARC position inside the WIVERN antenna pattern (Fig. 9), but only translates the received power at the ARC in time. The transmission time and flight time of



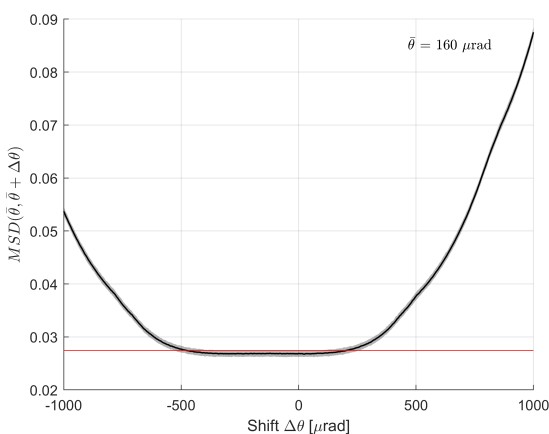
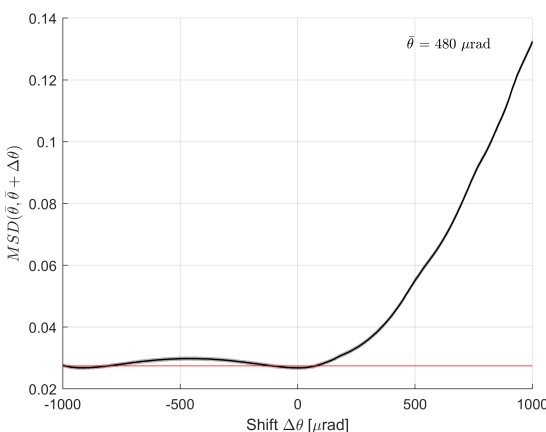

**Figure 10.** Example of least square distances (the continuous line corresponds to the 50th percentile, while the shading corresponds to the 5th and 95th percentile) for 400 different realizations of the antenna pattern with 1.0 dB of uncertainty as a function of the shift in elevation $\Delta\theta$ for an overpass with antenna boresigth passing at $\bar{\theta} = 160\,\mu$rad (left panel) and at $\bar{\theta} = 560\,\mu$rad from the ARC (right panel).

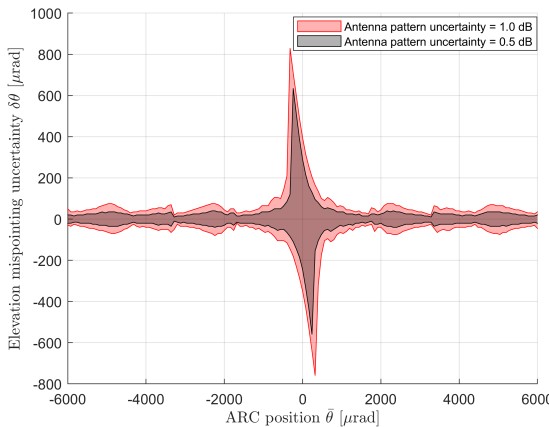
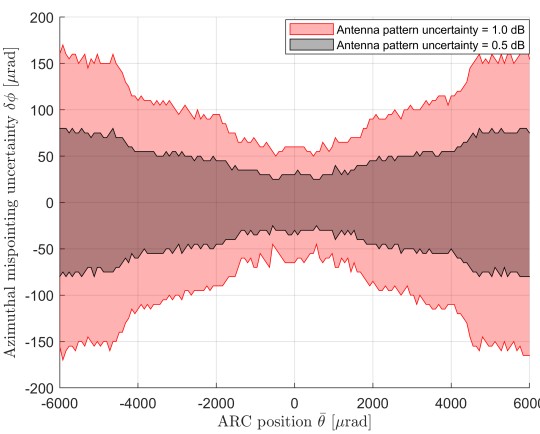

**Figure 11.** Uncertainty in the elevation (left panel) and azimuthal (right panel) mispointing determination as a function of the minimum ARC elevation distance, $\bar{\theta}$, from the WIVERN antenna boresigth. Cases with 0.5 (black) and 1.0 (red) dB uncertainty in the antenna pattern are shown.

the radar pulses ($\sim 2.2$ ms) are well known; excess path lengths in the atmosphere are expected to be less than few meters, thus delays are expected to be of the order of less than 0.01 $\mu$s, negligible in this context (Mangum and Wallace, 2015). The





procedure followed for the elevation mispointing is replicated introducing a shift in azimuth, $\Delta\phi$. In this case

$$MSD(\bar{\theta}, \Delta\phi) = \frac{\sqrt{\sum_{k=1}^{N_t} \left( P_{ARC}^{\bar{\theta}}(t_k) - P_{ARC}^{\bar{\theta}}(t_k + \Delta\phi/\Omega_a) \right)^2}}{N_t} \qquad\qquad -1000\,\mu rad \leq \Delta\phi \leq 1000\,\mu rad. \qquad (4)$$

As before, pdfs of MSD are computed and an estimate of the uncertainty in the azimuthal mispointing is derived based on percentiles. The right panel of Fig. 11 shows that the closer to ARC is the overpass, and the lower the uncertainty in the azimuthal mispointing determination is.

### 225   3.3.3    Expected number of useful calibration points as a function of ARC locations

The previous methodology requires the ARC to be positioned in a location within a few km (a few thousand $\mu$rad) from the radar boresight location at the ground. To estimate the number of useful calibration points as a function of ARC locations, the WIVERN orbit and boresight positions have been propagated for 50 days. Although the satellite ground track has a repeat cycle of 5 days, the boresight will not trace the same path after this period, thus different regions will be observed within the swath.

The simulation rationale consists of selecting a distribution of ARC locations over the region of interest (Fig. 12a) and counting the number of overpasses within a given footprint. A 1 km spacing in latitude and longitude has been selected to generate the ARCs distribution over the region, whereas a time step of 0.5 ms (equal to 250 m along the scan track) has been selected to guarantee a good sampling of the scan track. The simulation has been repeated considering the footprint corresponding to 1, 3, 5 and 10 times the antenna beamwidth.

Fig. 12 shows the average results for a 10 days period obtained from the 50 days simulation. Panel 12a displays the number of passes over the selected ARCs for a 10x beamwidth footprint. The image shows several hotspots located at specific latitudes and longitudes, while an optimal longitude-independent cluster of hotspots (red line) exists at around 79° of latitude. Since the enhanced number of passes at such locations is generated by the intersections occurring at the lower border of the swath, their positions are around 400 km south of the satellite ground track when it reaches its highest latitude. Three $50 \times 50\,\text{km}^2$ regions

have been selected around some of the hotspots at latitudes 45°, 66.5° and 78.7°. Panel 12b shows the histogram of the passes within these regions, considering the 10x beamwidth footprint. As expected, a greater number of passes occurs when moving towards higher latitudes.

Tab. 2 summarizes the results when taking different beamwidths. Clearly, when considering the 10x beamwidth footprint (i.e. within roughly 6000 $\mu$rad), a sensible number of passes (from at least more than 13 at 45° latitude to more than 53 at

78.7° latitude for a 10 days period) is possible over sites whose latitudinal position is properly selected.





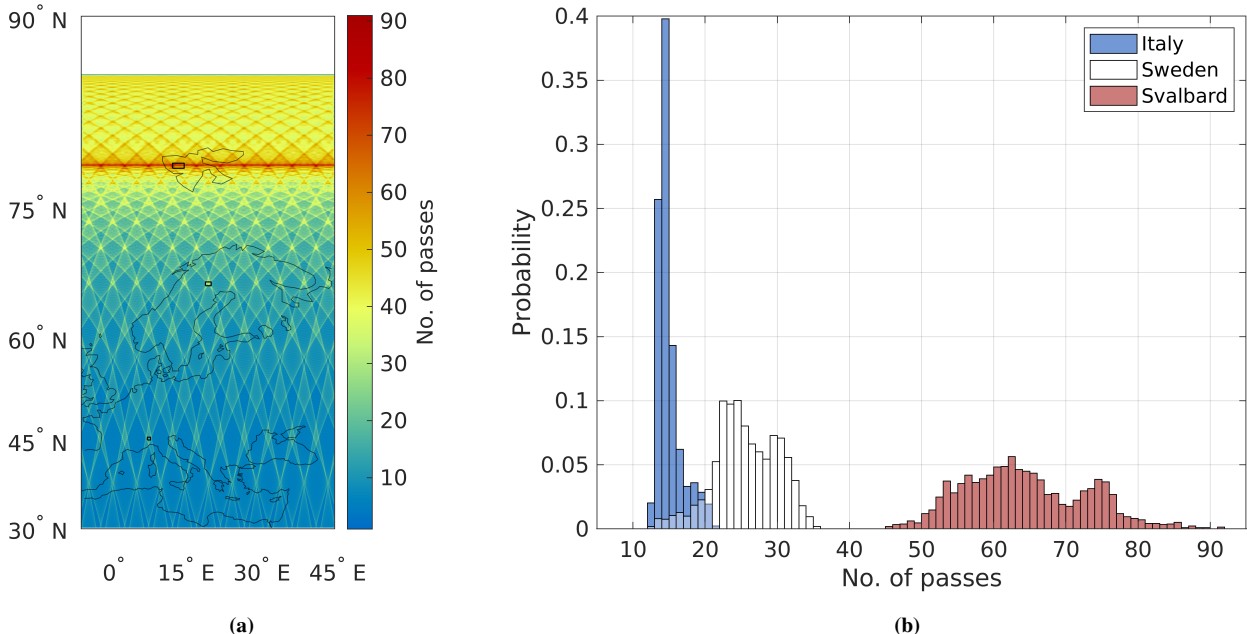

(a)           (b)

**Figure 12.** (a) ARCs average number of passes within the 10x beamwidth footprint over 10 days. (b) Histogram of the number of passes for the selected $50 \times 50\,\mathrm{km}^2$ regions (black rectangles in the left figure).

**Table 2.** 10th, 50th and 90th percentiles of the passes for the three selected regions over 10 days. The results refer to the passes within the footprints corresponding to 1, 3, 5 and 10 times the beamwidth.

|  | Italy | | | Sweden | | | Svalbard | | |
|---|---|---|---|---|---|---|---|---|---|
|  | 10th | 50th | 90th | 10th | 50th | 90th | 10th | 50th | 90th |
| 1x beamwidth | 1 | 1.4 | 2 | 1.4 | 2.4 | 3.6 | 4.6 | 6.2 | 8.2 |
| 3x beamwidth | 3.6 | 4.4 | 5.4 | 5 | 7.6 | 10.4 | 15.2 | 19 | 24 |
| 5x beamwidth | 6.6 | 7.2 | 8.6 | 9 | 13 | 16.8 | 25.9 | 31.6 | 39.2 |
| 10x beamwidth | 13.4 | 14.4 | 17.4 | 20.2 | 25.2 | 31.2 | 53.8 | 63.2 | 75.6 |



### 3.4 Correction method IV: ascending and descending orbit and ECMWF reference techniques

During a full rotation, the WIVERN instrument will look at the same LOS for azimuthal angles differing by $180°$. In such conditions, Eq. (1) predicts that the errors introduced by an azimuthal mispointing will be equal and opposite. But since these are errors on the LOS and the two directions here considered are opposite, it means that the two errors will be identical. Now,

for instance, in the part of the orbit closer to the equator, all winds observed at side views (i.e. with $\phi = 90°,\ 270°$ where the impact of the azimuthal mispointing is maximum) roughly correspond to the zonal winds. Then, in presence of an azimuthal mispointing that is changing at frequencies much lower than the orbital frequency (f=1.76×10$^{-4}$s$^{-1}$), the ascending and descending orbits will see opposite biases for the zonal winds (but this is true also for winds in any other direction, though the effect will reduce to zero when observing the meridional winds because of the $\sin\phi$ modulation). The advantage of side views

is also that at such angle the elevation mispointing is irrelevant (because of the $\cos\phi$ modulation). Therefore, in presence of a mispointing $\delta\phi$, the two pdfs of ascending and descending zonal winds collected at side views will be shifted by $\pm v_{SC}\sin\theta\delta\phi$ (i.e. the relative bias between ascending and descending orbits is about 1 m/s for $\delta\phi = 100\,\mu$rad).

     Statistically, after several orbits the two pdfs are expected to converge to the same pdf under the assumption that the zonal winds at local times differing by 12 h are the same. Is this assumption correct? In case, any discrepancy between the two pdfs

will be a signature of an azimuthal mispointing. But what is the sensitivity of this methodology? i.e. how long is it necessary to average in order to overcome the natural variability and what is the detectable bias for a given time scale?

     Alternatively, all H-LOS winds can be compared with ECMWF background forecast winds, that have been proved to be unbiased (biases are $\leq 0.3$ m/s in zonal wind and $\leq 0.15$ m/s in meridional winds), have good precision (standard deviations of the order of 2.5 m/s mostly because of unresolved small scale variability, Rennie (2022)) and have been exploited to correct

Aelous wind biases (Rennie et al., 2021). Each WIVERN H-LOS wind can be subtracted from the ECMWF reference. If WIVERN quality controlled winds are unbiased, then the distribution of the difference should have zero mean; otherwise, the bias could be estimated with an error which will be equal to $\sigma_{WIVERN-ECMWF}/\sqrt{N_{winds}}$ where $\sigma_{WIVERN-ECMWF}$ is the standard deviation of the distribution of the differences and $N_{winds}$ is the number of independent winds.

### 3.4.1 CloudSat-based analysis

To address these questions the dataset produced in Tridon et al. (2023) that combines the CloudSat reflectivity observations and the ECMWF winds has been exploited. Since CloudSat is orbiting in a sun-synchronous polar orbit like the one foreseen for WIVERN, the winds sampled in the ascending and descending orbits have the same statistical variability expected for WIVERN.

     Figure 13 shows an example of simulation of WIVERN measurements from a portion of CloudSat orbit through a North-

Eastern Atlantic widespread low which brought a significant amount of precipitation and high winds over Ireland on the 19$^{th}$ February 2007. The CloudSat reflectivity curtain (panel 13a) is combined with the corresponding ECMWF wind reanalysis (panel 13b) to simulate the slant reflectivity (panel 13c) and Doppler velocity (panel 13d) curtains that would be observed by WIVERN at side-view ($\phi = 90°$). Because of the slant incidence angle, the WIVERN surface reflectivity is much lower than

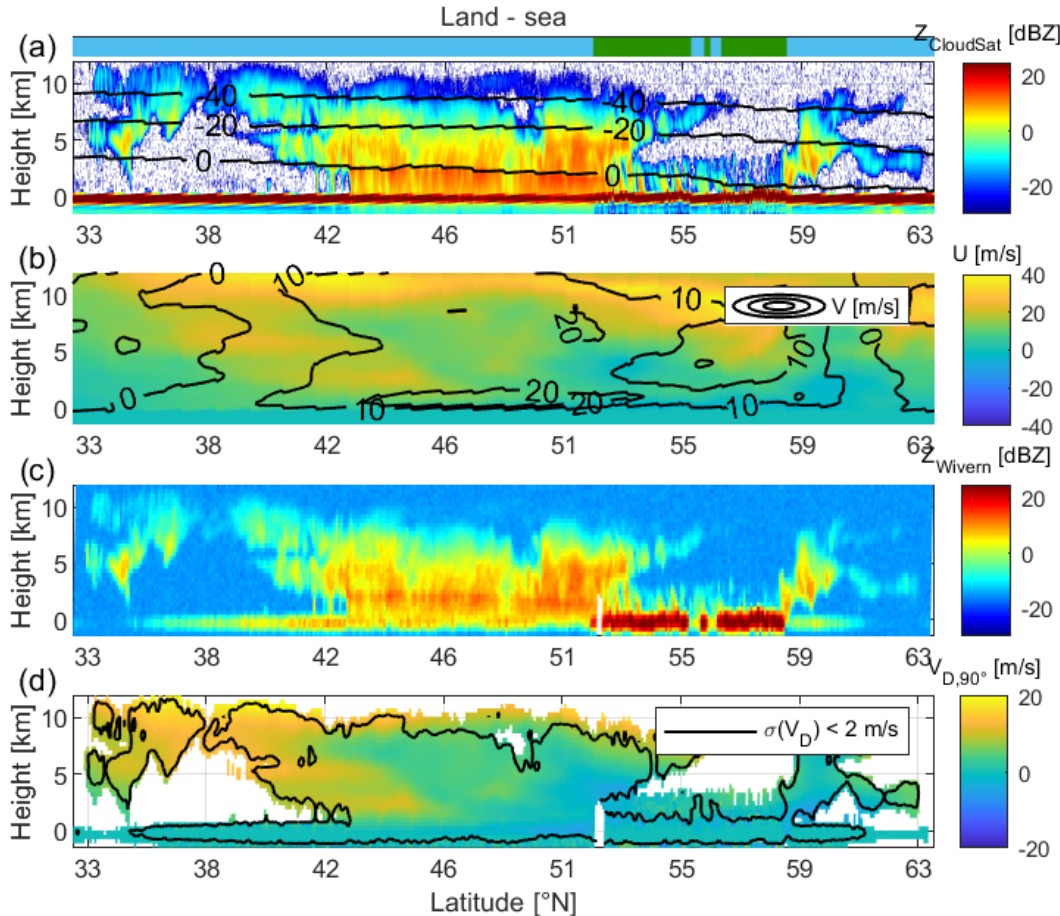

**Figure 13.** (a) CloudSat reflectivity of a frontal system over East Atlantic and British Isles, and ECMWF temperature contours. (b) Corresponding zonal (colors) and meridional (contours) ECMWF winds. (c) Simulated Wivern reflectivity at 10 km resolution. (d) Simulated side-view Wivern Doppler velocity at 10 km resolution with a contour showing areas with an accuracy associated to the radar estimator better than 2 m s$^{-1}$.

that of CloudSat apart from over land (see land/sea flag at the top of panel a). In panel 13d, the black contour highlights the
areas where the WIVERN Doppler velocity accuracy would be better than 2 m s$^{-1}$.

    Our method follows the following steps:

1. the dataset has been divided in ascending (A) and descending (D) orbits for latitudes between -65° and 65°;

2. histograms of WIVERN LOS winds when looking sideways to the right/left of satellite in A/D orbits (roughly corresponding to zonal winds) have been accumulated at different heights; only winds where clouds are present and produce


an SNR larger than -4 dB have been considered; random noise is added to each observation according to the expected
error computed from the radar simulator (formula 16 in Battaglia et al. (2022)) for integration lengths of 10 km.

3. ensemble of pdfs of WIVERN "zonal winds" are produced for different integration periods.

4. Jensen-Shannon distances (like in Battaglia et al. (2023)) between ascending and descending pdfs computed in 3 and for
A/D pdfs shifted by different wind biases (e.g. $\pm2$m/s corresponding to azimuth biases of 400 $\mu$rad).

5. threshold values expressed in terms of integration time or number of measured winds where different bias levels become
detectable for 4. are computed.

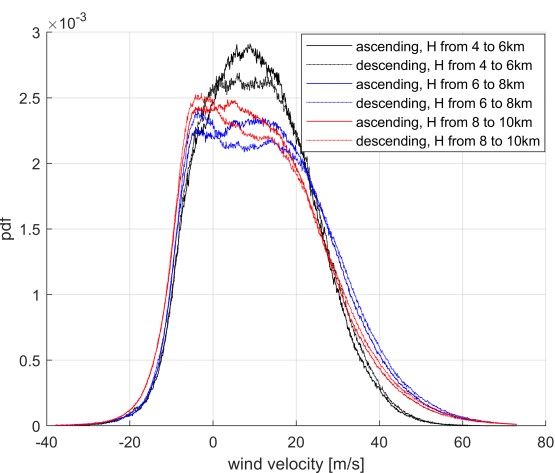
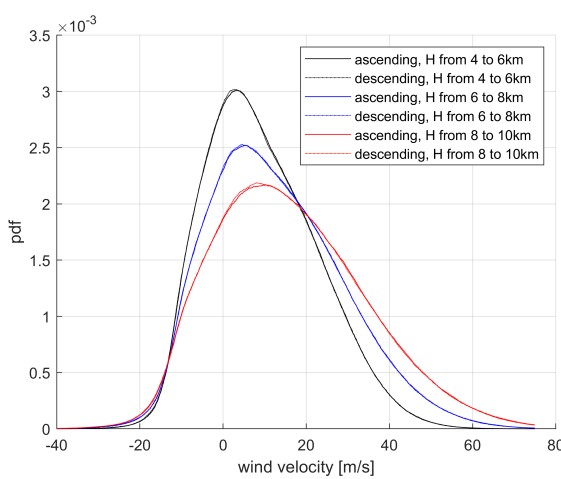

**Figure 14.** Pdfs of A (continuous line) and D (dotted line) in-cloud horizontal winds retrieved by Wivern when looking sideways (left panel) and all-sky horizontal winds at Wivern side view (right panel). In the left panel, only points characterized by a Doppler velocity accuracy better than 4 m/s are considered. The histograms have been generated with points sampled at latitudes within $\pm65°$ and at different altitude intervals, as indicated in the legend. The pdfs have been generated with 270 (in-cloud winds) and 365 (all-sky winds) days of data.

Pdfs of in-cloud horizontal winds retrieved by WIVERN when looking sideways are shown in the left panel of Fig. 14 for A/D orbits (dotted/continuous) and for different height ranges as indicated in the legend. The same plot is repeated on the right hand side considering all-sky winds. In the latter condition, A and D wind distributions looks pretty much identical. However,

when considering only in-cloud winds, the two distributions take different shapes, suggesting the existence of a diurnal cycle (A and D winds are sampled twelve hours apart) affecting in-cloud winds. This result makes the option of identifying azimuthal mispointing only by using WIVERN ascending and descending measurements challenging because the pdfs of A/D in-cloud winds are intrinsically different and therefore large biases in the winds (typically of the order of 2 m/s) are needed to see a neat separation between the two pdfs for accumulation times of at least 10-15 days (not shown).





On the other hand, when considering the ECMWF as a reference, Fig. 15 demonstrates that the histogram of the differences for the side winds has a standard deviation of the order of 3.66 m/s with an average of about 80,000 winds per day. In this case, only WIVERN's wind measurements characterized by a SNR higher than 0 dB have been taken into account. Because of this large amount of winds, this demonstrates that the error on the estimate of the azimuthal bias will become negligible ($\leq 0.2$ m/s) already after few minutes. The real limit becomes in this case the assumption that the reference ECMWF winds are unbiased.

The validity of such assumption applies on global averages over few days with an upper limit for the bias of circa 0.3 m/s. Note that the same reasoning can be applied to the forward/backward line of sight winds and therefore to the elevation biases.

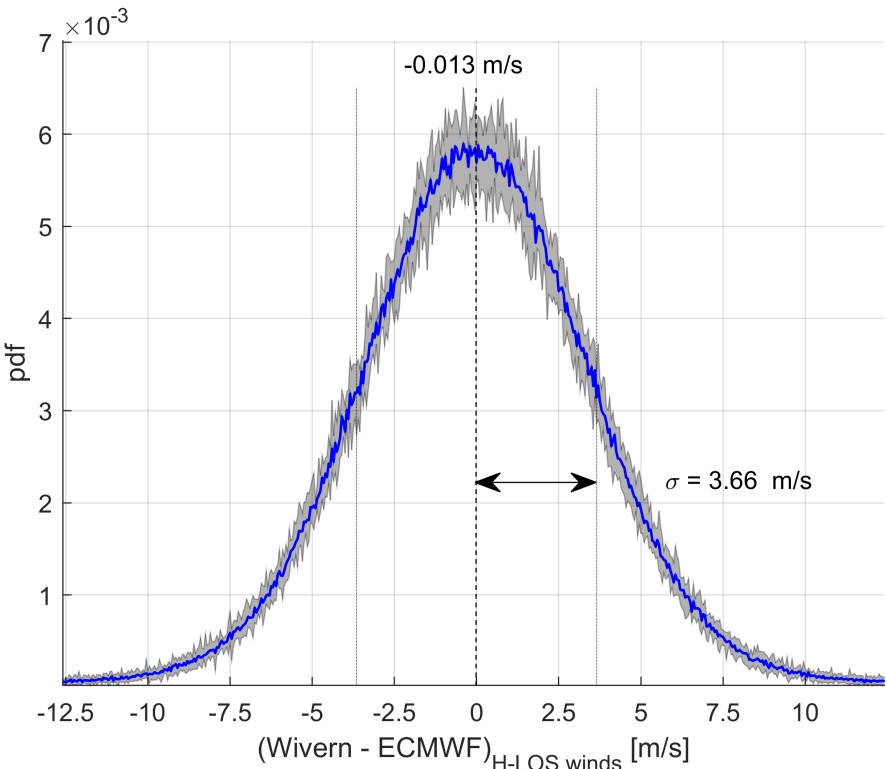

**Figure 15.** Pdf of the difference between the winds retrieved by Wivern (LOS) at side view and the ECMWF winds. The pdf have been generated with 801,474 points, all with an SNR larger than 0 dB, collected during a period of 10 days (blue line). The envelope of the one day pdfs collected in each of those 10 days is also shown (gray shading).





## 4 Summary and conclusions

Different methodologies for correcting mispointing errors in conically scanning Doppler measurements (with focus at the WIVERN configuration, currently under study as one of the Earth Explorer 11 candidate missions) have been discussed. Results show that:


- The use of radar in "altimeter" mode is very robust for identifying elevation mispointing on very short time scales (few ms). Depending on the surface peak strength and the integration length, different levels of correction can be achieved; e.g. with a 10 dB peak to noise ratio less than 20 $\mu$rad (50 $\mu$rad) can be achieved at 10 km (2 km) integration length. This value corresponds to velocity errors smaller than 0.12 m/s (0.28 m/s). The methodology is limited by the flat surface assumption and by the absence of low atmospheric targets that may contaminate the surface signal. Proper screening to identify these situations must be performed before hand.


- The surface Doppler profile can be used for correcting both elevation and azimuthal mispointing but with generally worse performances than the previous method. With a 10 dB surface peak to noise ratio errors of 0.4 m/s (0.9 m/s) can be achieved at 10 km (2 km) integration length. The method is likely to produce accurate pointing corrections when making use of the clear sky, high peak to noise ratio flat surfaces encountered across several antenna rotations. Limitations similar to the previous method apply in this instance; additionally, for ocean surfaces, the potential bias introduced to the Doppler by waves and currents must be accounted for.


- The use of an active radar calibrator is effective in identifying slowly changing mispointing errors (biases) larger than about 50 $\mu$rad when considering multiple overpasses over week-long periods with the error mainly driven by the knowledge of the details of the antenna pattern. The location of the ARC can be optimally chosen based on the orbit details, with the goal of maximising the number of overpasses.


- Winds measured by WIVERN in ascending and descending orbits can be used to detect azimuthal biases but only for large biases (of the order of 200 $\mu$rad) on time scale longer than 10 days. On the other hand, because of the huge number of WIVERN winds measurements collected every day, the comparison of the Level 2 WIVERN HLOS wind to a state-of-the-art data assimilation system and forecast model like provided by ECMWF via the so-called O-B (observation - background) technique is extremely effective in determining biases. The method is practically only limited by time and spatial scales at which the reference model can be considered unbiased.


Future work should address the impact of the instrument footprint variability of the surface $\sigma_0$ (e.g. due to differential attenuation or by surface height variability) for methodology 1 and 2. The impact of waves and currents on the Doppler measurements should also be established at this high frequency and at slant incidence angles.


*Author contributions.* FES performed most of the simulations and the analyses. AB wrote most of the text and has defined the project. FT performed the analysis on the statistics of useful surface return and provided the simulations for the CloudSat based analysis. PM performed





the analysis on the expected number of passes on different ARC locations. RD performed the analysis on the flat surface approximation. AI contributed to the discussion and the review of the paper.

*Competing interests.* The authors declare that they have no conflict of interest.

*Acknowledgements.* This work was supported in part by the European Space Agency under the activity WInd VElocity Radar Nephoscope (WIVERN) Phase 0 Science and Requirements Consolidation Study, ESA Contract Number 4000136466/21/NL/LF. AB's work was funded by Compagnia di San Paolo. FES's work was conducted during and with the support of the Italian national inter-university PhD course in Sustainable Development and Climate change (link: www.phd-sdc.it). This research used the Mafalda cluster at Politecnico di Torino.



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
