# Peer review of "Mispointing characterization and Doppler velocity correction for the conically scanning WIVERN Doppler radar"

_Atmospheric Measurement Techniques, 2023_

## Author Comment (AC1)

We thank the reviewer for his review and his detailed comments.

Below you will find the reviewer's comments in bold and our replies.

**1 Introduction**

**L 16: Abbreviation ALADIN not introduced.**

We will add the meaning of this abbreviation.

**Please revise the citations for the Aeolus mission. E.g. Lux et al., 2021, is a paper on a very specific topic on laser frequency stability and probably not a good reference for the whole Aeolus mission. A good overview of the Aeolus mission is given e.g. by ESA (2008).**

We revised the citations related to the Aeolus mission. As suggested, we added ESA (2008) as a reference for the Aeolus mission.

**In line 24f, you motivate the WIVERN radar as a complementation of current global wind measurements in cloudy conditions. However, you do not clearly state the limitations of the other observation techniques, especially Aeolus, before. This could be added e.g. in line 21.**

Two sentences have been added to the paragraph to add context.

Aeolus only measures winds in clear sky (so called ``Rayleigh winds'') and at inside thin clouds (``Mie winds''). Scatterometer measurements can also provide complementary observations at the surface, with significant progress being recently achieved even in presence of strong winds near heavy rain (Polverari, 2022). Similarly, winds at cloud top can be derived from successive satellite images of clouds and humidity, the so-called Atmospheric Motion Vectors. However, apart from sporadic and sparse radio soundings and aircraft penetrations, no wind observations are currently available inside thick clouds and precipitating systems.

**2 Mispointing errors**

**Your whole paper is based on equation (1). Could you please add a proper derivation of this very important equation?**

We will add a better derivation in the manuscript. The hydrometeor Doppler velocity, $v_D$, is obtained by subtracting the component of the spacecraft velocity, $v_{SC}$, along the antenna boresight form the measured Doppler velocity, $v_{mD}$. Thus, $v_D = v_{mD} - v_{SC} \sin(\theta) \cos(\varphi)$, where $\theta$ and $\varphi$ are the elevation and azimuthal pointing angles, respectively, and with the second term of the equation representing the projection of the spacecraft velocity along the antenna boresight. If the actual pointing of the antenna has a mispointing of $\delta\theta$ and $\delta\varphi$ in the elevation and azimuthal angle respectively, the mispointing error will be: $\delta v_{mis} = [v_{mD} - v_{SC} \sin(\theta) \cos(\varphi)] - [v_{mD} - v_{SC} \sin(\theta + \delta\theta) \cos(\varphi + \delta\varphi)]$

$$\frac{\delta v_{mis}}{v_{SC}} = [\sin(\theta + \delta\theta) \cos(\varphi + \delta\varphi) - \sin(\theta) \cos(\varphi)]$$
$$= \left[ (\sin(\theta) + \cos(\theta)\delta\theta)\big((\sin(\varphi) + \cos(\varphi)\delta\varphi)\big) - \sin(\theta) \cos(\varphi) \right]$$

$$\delta v_{mis} = v_{SC}[-\sin(\theta) \sin(\varphi)\, \delta\varphi + \cos(\theta) \cos(\varphi)\, \delta\theta]$$

**3 Doppler correction methods**

**Eq (2): The variable $H_{S/C}$ is not introduced.**

$H_{S/C}$ is the spacecraft altitude. We added the meaning of it in the text. Also changed notation to $H_{SC}$.

**L 79: Why "therefore"? Maybe better "unfortunately"?**

We agree with your suggestion, and we replaced "therefore" with "unfortunately".

**L 87: blue -> orange dashed**

Corrected.

**L 88: How is the noise subtracted? Does this noise subtraction method have an impact on the precision of your correction method(s)?**

The noise subtraction is via a standard procedure (e.g. description in Kollias et al. 2022, added). Here we have first simulated noise according to the Wivern spec (and add to the signal generated by the surface), then the noise level is estimated by averaging clear sky bins; finally, this estimate is subtracted from the total signal.

**L 140: Why 0.4 m/s? In the introduction you state that it should be below 0.3 m/s.**

In the introduction we are saying that previous studies have considered mispointing errors negligible (i.e. lower than 0.3 m/s). This is not the case anymore. Additional text has been added to explain what the current requirements are. In order to fulfill the mission requirements, when counting for the other contributions (pulse pair estimator error, non uniform beam filling, wind shear) the pointing contribution of the random LoS Doppler velocity error budget must be of the order of 0.4-0.6 m/s whereas the requirement for the systematic contribution has to be smaller than 0.3-0.6 m/s.

**L 151f: Why do you choose these box sizes?**

Because they correspond roughly to from (500 m x 500 m) to (500 m x 5 km), which are the instantaneous WIVERN footprint and the area after 5 km integration, respectively.

**Figure 7: What is the difference between dashed and solid lines? Description is missing.**

The solid lines are referred to the forward pointing case (azimuthal angle = 0 degrees), while the dashed lines are referred to the side view pointing cases (azimuthal angle = +-90 degrees). We will add the description in the caption of the figure.

**L265: The method is in detail described by Weiler et al. 2021. So maybe this is a better reference here.**

We replaced the reference as you suggested.

**Figure 13: Is this Figure really necessary?**

Yes, we think that this figure is important because it explain how winds can be reconstructed and properly sampled in in-cloud region with reflectivities exceeding about –15 dBZ from the CloudSat database.

**L 182: Why only between -65° and 65° latitude?**

Since we want to compare the winds retrieved at side views during the descending orbits with the ones retrieved at side views during the ascending orbits, we preferred to exclude latitude close to the orbital inclination because those latitudes mark a transition for side LoS winds from zonal to meridional winds.

**Figure 14: Difficult to see the difference between solid and dotted lines. Please adapt line style.**

All lines are solid but have different markers now.

**4 Summary and conclusion**

**L 332: And on which time scales can such models be considered unbiased? What about the wish to have model-independent measurements?**

In our study we used ECMWF, which can be considered unbiased at global scale probably on weekly time scales. Comparison with radiosoundings and aircrafts suggest expected bias <0.3m/s with standard deviations around 2.5m/s.

**In the introduction you motivate your study with phase 0 industry studies showing mispointing errors above 0.3 m/s particularly for slow varying components. Are the time scales of your methods sufficiently fast to correct for these? So, what exactly are "slow varying components"? This should be addressed either here or in the introduction.**

Different error contributions have been classified and assigned to systematic or random errors according to the split frequency of 1.16e-5 Hz, which corresponds to 1 day period. Thus, "slow varying components" are the mispointing power spectral density components characterized by a period larger than 1 day. Time scales of method #1 and #2 are enough fast to correct these errors. Method #3 is slower than the previous two and can be used to correct slowly changing mispointing errors. Time scale of method #4 is driven by the time scale at which the reference model can be considered unbiased; in our analyses we used the model provided by ECMWF and it looks very effective to correct these errors. We will address this in the introduction.

---

## Author Comment (AC2)

**Reply to Reviewer 3**

We thank the reviewer for his review and his detailed comments.

Below you will find the reviewer's comments in bold and our replies.

**General comments:**

**I recommend to add "velocity" after each occurrence of "Doppler" to increase readability of the text. Also make sure that whenever you draw a conclusion that is based on Eq. 1, state that in the appropriate spot of the description of Doppler velocity correction method 1-4.**

We will add what you suggested.

**Also, in terms of structure of manuscript, consider removing the numbering of subsubsections 3.1.1. and 3.2.1 as there are no further subsections (3.1.2 and 3.2.2).**

We will remove them in the revised version.

**Minor comments:**

**Line 2: congratulations to making it to the next phase – replace "four" by "two" candidates for ESA's Earth Explorer 11 mission**

Thank you. We will replace "four" by "two" in the revised version.

**Line 4: add "Doppler" in front of "velocities"**

We will add it.

**Line 19: define acronym ALADIN**

We will add its definition in the revised version.

**Line 20: define acronym NWP**

We will add its definition in the revised version.

**Line 26 – 27: Rephrase this convoluted sentence, e.g. as "With clouds covering roughly 30% of the tropospheric volume, Doppler cloud radars have the potential to complement wind observations by Doppler lidar in clear-sky and thin cloud conditions".**

We will rephrase that sentence as you suggested.

**Line 32: be more precise about the "large spacecraft velocity": add velocity range in brackets**

Typically, in LEO, the orbital velocity of the satellite is of the order of 7.6km/s, we will specify it in the text.

**Line 42: "sources" (not source)**

Corrected.

**Line 59: after "Doppler mispointing error" add " in Doppler velocity deltav_mis"**

We will add it.

**Line 67 – 69: Expand this short paragraph by stating which technique is applicable to azimuth- or elevation mispointing or both or alternatively mention this fact at the subheadings of each technique (e.g. in line 70: …"elevation mispointing correction of Doppler velocities")**

We will expand it specifying what technique is applicable to what mispointing type.

**Line 75: define acronym AOCS**

Done.

**Line 103-104: Add "According to Eq. 1" in front of "the last three solutions…"**

Added.

**Line 127 – 128: Clarify what you mean by "surface Doppler at all heights".**

We mean "surface Doppler velocity at all range gates / height bins". We will clarify it in the text.

**Line 138: Do you mean "corrected" instead of "achieved"?**

We mean "of the uncertainty in the velocity error correction that can be achieved with this methodology". We will clarify it in the text.

**Line 163: Replace "looks" with "views"**

Done.

**Line 223: To increase readability, rephrase to "…the closer the overpass is to ARC, the lower is the uncertainty in the azimuthal mispointing determination"**

Rephased as you suggested.

**Line 259: Add "this" in front of "case"**

Done.

**Comments on Figures:**

**Fig 5:  Where are the thin dotted black lines?**

The thin dotted black lines are the isodops lines. We will modify their style in order to increase the readability of those lines.

**Fig 7: What do solid and dashed lines refer to?**

Solid and dashed lines are referred to forward pointing (azimuthal angle = 0 degrees) and side pointing (azimuthal angle = +-90 degrees) cases, respectively. We will add this description in the caption of the figure.

**Fig 10 + 11: increase font size of axes and labels**

We will increase the font size of axes and labels in the revised version.

**Fig 11: right panel: captions says theta = 560 microrad, legend (and text) state theta = 480 microrad.**

Theta = 480 microrad is the correct value. We will correct the captions.

**Fig. 12 and Table 2: replace "passes" with "overpasses"**

We will replace it in the revised version.

---

## Author Comment (AC3)

We thank the reviewer for his review and his detailed comments.

Below you will find the reviewer's comments in bold and our replies.

**This paper examines four methods for diagnosing pointing errors for the proposed WIVERN mission, which would use a Doppler radar performing a conical scan for wind measurements in cloudy areas. I think the title and introduction give the impression that errors are being corrected. However, I found the focus of the paper to be a little different, namely focusing on the errors in estimating the mispointing but not necessarily addressing the problem of correcting the pointing and the associated velocity. I think evaluation of the mispointing is definitely worthwhile, but the authors may want to consider adjusting the title and introduction. My other concern is that the nature of the pointing errors doesn't seem completely general. From Figure 1, it looks like the rotation axis is assumed vertical so that the only errors are a deviation from the elevation angle theta and a deviation of the azimuth angle from that estimated. This means that that the azimuth and elevation angles appear to be considered independently here. Should errors in which azimuth and elevation are coupled also be considered? Could such errors occur, as with a scan-axis mounting offset? Perhaps this is already included but it wasn't clear to me. Lastly, I think adding some additional math to describe each method may make the work considerably clearer.**

We agree on the comment, the title is misleading, we will modify it as follow: "Mispointing characterization and Doppler velocity correction for the conically scanning WIVERN Doppler radar". The methods described in the paper do not correct the mispointing itself. In fact, for the mission, it is not paramount to have an accurate and precise pointing, but it is essential to have pointing knowledge within tens of microradians in order to correctly subtract the satellite velocity component along the antenna boresight from the measured Doppler velocity to retrieve the hydrometeor Doppler velocity. The methods quantify the mispointing angles and estimate the error induced by such mispointings on the Doppler velocity. Once the error on the Doppler velocity has been estimated based on Eq. 1, the Doppler velocity is then corrected. Yes, we assume that the azimuthal and elevation mispointing errors are independent one from the other, and further studies must be performed to better evaluate combinations of the two mispointing errors.

$A_{roll}$, $A_{pitch}$ and $A_{yaw}$ are the roll, pitch and yaw angles, respectively, that characterize a scan-axis mounting offset. If they can be assumed to be constant over time, they will induce the following biases in the azimuthal ($e_\phi$) and elevation ($e_\theta$) direction:

$$e_\theta(t) \simeq [-sin(\Omega * t) \quad cos(\Omega * t)] * \begin{bmatrix} A_{roll} \\ A_{pitch} \end{bmatrix}$$

$$e_\varphi(t) \simeq [-cos(\Omega * t) \quad -sin(\Omega * t) \quad 1] * \begin{bmatrix} A_{roll}/tan(\theta) \\ A_{pitch}/tan(\theta) \\ A_{yaw} \end{bmatrix}.$$

If $A_{roll}$, $A_{pitch}$ and $A_{yaw}$ are unknown and are assumed to be constant (e.g. due to the presence of a roll, pitch and/or yaw mispointing error caused by post-launch conditions), they can be retrieved using method #1 ($A_x$ and $A_y$) and #2 ($A_z$). In this case, $\delta z$, (the difference between the altitude at which the surface reflectivity peak is located and the altitude at which we expect the peak is located) can be averaged over a period enough long (with respect to the azimuthal and elevation mispointing error frequency) so that the effect on $\delta z$ given by the elevation ($\delta\theta$) and azimuth ($\delta\phi$) mispointing is averaged to zero. Then, $A_{pitch}$ can be retrieved by looking at the $\delta z$ at the forward and the backward view. $A_{roll}$ can be retrieved by looking at the the the $\delta z$ at right side and at left side view. $A_{yaw}$ can be handled as a $\delta\phi$ error and retrieved with method #2. If the reviewer thinks this should be added to the manuscript, we will include this in an appendix.

**Equation (1): Please include a reference for this equation or a derivation, either in the text here or as an appendix. This would help with the concern about coupling between azimuth and elevation errors.**

We will add the derivation in the text. The hydrometeor Doppler velocity, $v_D$, is obtained by subtracting the component of the spacecraft velocity, $v_{SC}$, along the antenna boresight from the measured Doppler velocity, $v_{mD}$. Thus, $v_D = v_{mD} - v_{SC} \sin(\theta) \cos(\varphi)$, where θ and φ are the elevation and azimuthal pointing angles, respectively, and with the second term of the equation representing the projection of the spacecraft velocity along the antenna boresight. If the actual pointing of the antenna has a mispointing of δθ and δφ in the elevation and azimuthal angle respectively, the mispointing error will be:

$$\delta v_{mis} = [v_{mD} - v_{SC} \sin(\theta) \cos(\varphi)] - [v_{mD} - v_{SC} \sin(\theta + \delta\theta) \cos(\varphi + \delta\varphi)]$$

$$\frac{\delta v_{mis}}{v_{SC}} = [\sin(\theta + \delta\theta) \cos(\varphi + \delta\varphi) - \sin(\theta) \cos(\varphi)]$$

$$= [(\sin(\theta) + \cos(\theta)\delta\theta)(\cos(\varphi) - \sin(\varphi)\delta\varphi) - \sin(\theta) \cos(\varphi)]$$

$$\delta v_{mis} = v_{SC}[-\sin(\theta) \sin(\varphi)\,\delta\varphi + \cos(\theta) \cos(\varphi)\,\delta\theta]$$

**Line 65: I would expand the phrase in parentheses for better clarity, for example changing i.e., to "is". Even more clear would be "forward direction is phi=0; backward is phi=pi"**

We will expand it as you suggested.

**Line 91: insert ",it " after "threshold"**

We will correct it.

**Line 100: For this technique, the delta-z is translated to a delta-theta via (2). The delta-z error comes from mis-estimation of the peak of the surface return, as shown in Figure 2. Are there additional uncertainties in apply (2) that would increase the delta-theta error, e.g., timing jitter?**

Any phenomenon that might cause uncertainty in range determination, such as the timing jitter, will cause higher uncertainty in determining the delta-theta. But such uncertainties are assumed negligible in this study.

**Line 125: I suggest expanding a bit on the need for flatness. While directly affecting the range to the surface, for method I, its effect on Doppler is not obvious. How would a surface with non-moving topography affect the measured Doppler? How much of the Earth's land would qualify as flat?**

Figure 8 shows how much of the Earth's land can be considered flat based on different criteria. We are currently implementing a software to compute the surface reflectivity and Doppler return in presence of mountainous regions. But this is out of the scope of the present study.

**Figure 7: what are the dashed curves?**

The solid lines are referred to the forward pointing case (azimuthal angle = 0 degrees), while the dashed lines are referred to the side view pointing cases (azimuthal angle = +-90 degrees). We will add this description in the caption of the figure.

**Line 167: In receiver mode, I assume that the ARC does not return a signal to the radar. If so, then this analysis uses the data recorded by the ARC. Further, my understanding is that the ARC will get scanned only once, so the example in Fig 9b is the data that would be recorded by the ARC on a single sweep by the**

radar antenna. Hence, the ARC essentially records a cut through the 3D pattern. Due to the circular shape, the cut is slightly bent, as noted in the text. Is this a correct description of the ARC measurement? Assuming so, my further understanding is that for an azimuth error, there is a resulting time offset. Are there other, non-pointing, factors that could also cause errors, e.g. timing? What about ARC geolocation accuracy? Are there other error factors in the elevation measurement?

Yes, this is a correct description of the ARC measurement. Depending on different situations, multiple sweeps per overpass can be recorded by the ARC; however, we assumed that only one sweep would be recorded per overpass. Yes, there are some other factors that affect the accuracy of the ARC methodology to detect and correct the mispointing velocity biases. In our analysis, we assume that we perfectly know the geolocation of the ARC, the position of the spacecraft, the propagation time of the signal and the time at which the WIVERN radar send the pulses. Uncertainty in the knowledge of those factors translates in a further uncertainty in the detection and correction of the mispointing Doppler velocity error. We will specify this in the revised version.

**Line 249: By "identical", do the authors mean equal magnitudes but opposite signs?**

We mean equal in magnitudes and equal in signs. When WIVERN is looking at azimuthal angles differing by 180 degrees, the two ensembles of LOS wind velocities would be equal in magnitude and opposite in sign. From equation (1) we see that also the LOS wind velocity errors are equal in magnitude and opposite in sign. Thus, the errors in the actual wind velocity retrieved (not LOS) will be identical in magnitudes and signs. We will specify that in line 249 we are referring to the error on the actual wind velocity.

**Line 259: My understanding is that this paragraph means that the A and D pdfs will converge to one if there is no azimuth error. However, if the two pdfs differ due to the azimuth error, the difference would continue with more data. is this correct - if not, please clarify. I think some minor editing to improve the clarity would be good. Also, I think the frequency of the error needs more explanation. If the frequency is much higher (faster) than the orbital frequency, then presumably both A and D pdfs are smeared; clarification would help. Instead of varying quickly, what if the bias is constant? I think showing equations for the winds and biases would make this much clearer.**

The A and D methodology is applicable for azimuthal biases which are constant during very long time scales (e.g. monthly), thus, which remains constant along multiple orbits (i.e. their frequency is lower than the orbital frequency). Instead, if the error frequency is much higher than the orbital frequency, it means that the error would not be constant during several orbits, and this method is not enough fast to tackle it.

The crucial assumption for the methodology to work is that there is no diurnal cycle of in-cloud zonal winds. However, this hypothesis has been rebutted when looking at the ECMWF winds co-located with CloudSat CPR detected clouds. Therefore, this method is not applicable except for detecting large biases after long times. Yes, your statements on the A and D pdfs are correct. We didn't show equations for the winds and biases because this methodology does not work. However, the same kind of methodology has been used in the paper *Battaglia, A., Scarsi, F. E., Mroz, K., and Illingworth, A.: In orbit cross-calibration of millimeter conically scanning spaceborne radars, Atmospheric Measurement Techniques, 16, 3283–3297, https://doi.org/10.5194/amt-16-3283-2023, 2023* and we will add the reference to it.

**Line 294: Please better define "all-sky" - is this all directions, instead of just sidelooking?**

We meant "all-sky conditions". We will specify it in the text.

**Line 305: The results of Figure 15 look promising. However, I'm not clear on how the statistics here translate to an uncertainty in the azimuth offset. An uncertainty of 200 urad is provided in the conclusions, but it's not clear where that came from. Please add a short discussion on how the 200 urad can be derived from Figure 15.**

The uncertainty of 200 urad is not derived from Figure 15. Figure 15 and 200 urad uncertainty are related to two different techniques. Section 3 describes two techniques to detect and correct the azimuthal errors. The first technique (which does not work) is based on comparing the horizontal LOS winds at side views retrieved during the ascending orbit with the ones retrieved during the descending orbit. This technique has an uncertainty of 200 urad caused by the fact that there is a diurnal cycle in the in-cloud zonal winds (see Figure 14). So, 200 urad is the uncertainty related to the *ascending and descending orbit* technique.

The second technique consists in comparing the WIVERN H-LOS winds sampled at side views with the ECMWF winds. The pdf of the difference between these two winds is shown in Figure 15.